# Laser Irradiation of a Bio-Waste Derived Carbon Unlocks Performance Enhancement in Secondary Lithium Batteries

**DOI:** 10.3390/nano11123183

**Published:** 2021-11-24

**Authors:** Mariangela Curcio, Sergio Brutti, Lorenzo Caripoti, Angela De Bonis, Roberto Teghil

**Affiliations:** 1Dipartimento di Scienze, Università della Basilicata, Viale dell’Ateneo Lucano 10, 85100 Potenza, Italy; angela.debonis@unibas.it (A.D.B.); roberto.teghil@unibas.it (R.T.); 2Dipartimento di Chimica, Università di Roma La Sapienza, Piazzale Aldo Moro 5, 00185 Roma, Italy; caripoti.1765693@studenti.uniroma1.it; 3GISEL—Centro di Riferimento Nazionale per iSistemi di Accumulo Elettrochimico di Energia, INSTM Via G. Giusti 9, 50121 Firenze, Italy

**Keywords:** hard carbons, laser irradiation, lithium-ion batteries, negative electrodes, bio-waste materials

## Abstract

Pyrolyzed carbons from bio-waste sources are renewable nanomaterials for sustainable negative electrodes in Li- and Na-ion batteries. Here, carbon derived from a hazelnut shell has been obtained by hydrothermal processing of the bio-waste followed by thermal treatments and laser irradiation in liquid. A non-focused nanosecond pulsed laser source has been used to irradiate pyrolyzed carbon particles suspended in acetonitrile to modify the surface and morphology. Morphological, structural, and compositional changes have been investigated by microscopy, spectroscopy, and diffraction to compare the materials properties after thermal treatments as well as before and after the irradiation. Laser irradiation in acetonitrile induces remarkable alteration in the nanomorphology, increase in the surface area and nitrogen enrichment of the carbon surfaces. These materials alterations are beneficial for the electrochemical performance in lithium half cells as proved by galvanostatic cycling at room temperature.

## 1. Introduction

To tackle the simultaneous challenge of the massive need of natural resources for industrial transformations and the enormous accumulation of solid waste from human activities, a transition from a linear to a circular economy is inevitable. In this direction, innovative transformative processes designed to reuse waste can save nonrenewable resources and pave the way for a sustainable global economy [1,2].

For example, the conversion of bio-waste to high added-value carbon-based materials such as hard carbons can be the key for the realization of sustainable negative electrodes for secondary batteries [3].

Nowadays most of portable devices and electric vehicles are powered by rechargeable Li-ion batteries (LIBs). Conventional negative electrodes are based on graphite or nanographitic carbons which are able to give highly reversible Li intercalation/deintercalation in batteries. Despite its commercial success, graphitic carbons have a limited theoretical capacity of 372 mAhg^−1^ and an even smaller practical capacity, i.e., <350 mAhg^−1^, and therefore great attention has been devoted since the 90s to low-crystalline/disordered carbons, which can sustain larger lithium intake/outtake, good reversibility and tunable nano-morphologies depending on the preparation conditions. [4,5].

In recent years many research groups worldwide have been developing innovative sustainable processes to obtain innovative carbonaceous materials for LIBs. Starting with bio-waste from food industry and agriculture [4,5,6,7,8,9,10,11,12], they investigated the impact of the nature of bio-waste on the resulting porous carbons and their performances in LIBs. Carbons obtained from natural sources can be grouped in lignin-, hemicellulose- and pectin-based bio-wastes, according to the more abundant natural polymer in the pristine material. Carbons obtained from lignin-based bio-waste show the best electrochemical performances in term of specific capacity, capacity retention and coulombic efficiency compared to carbonaceous materials from other natural matrixes. The pyrolysis of lignin, in fact, leads to an optimal microporosity in the resulting carbon, positively enhancing its ability to store Li-ions [8,13].

Different lignin-rich raw materials have been studied as carbon precursors for LIBs negative electrodes focusing on highly abundant sources, such as peanut shells [14], banana fibers [4,13], hazelnut shell [5], wheat stalk [9], coir pith [8], corn stalk [11], and rice husk [10]. Overall, the electrochemical properties of waste-derived carbons depend not only on the waste source but are also strongly impacted by the preparation. To this aim, pyrolysis at a high temperature can be coupled with simultaneous porogenic processes using sodium or potassium hydroxide activation [8,10,11,13]. Among the very large literature, for example, Unur et al. [5] produced microporous carbons by hydrothermal carbonization (HTC) of hazelnut shells and investigated the effect of three different porogenic treatments: simple heat treatment, potassium hydroxide activation, and magnesium oxide (MgO) templating. In this way, they demonstrated that sole heat treatments on samples prepared by HTC enable better electrochemical performances as anode materials for LIBs thanks to the optimal balancing of nano-graphitic ordering, large surface area and effective micro-porosity.

In this study we demonstrate the beneficial effect of the post-pyrolysis irradiation with a pulsed laser of a hard carbon derived from hazelnut shells. Our goal is to enhance the performance of a hazelnut-shell derived hard carbon as active material in negative electrodes for LIBs.

Hazelnuts are heavily used in the food industry and are produced worldwide mainly in the Mediterranean area; Italy and Turkey being the major world producers [15]. Hazelnut shells are largely constituted by lignin, cellulose, and hemicelluloses, with different percentages depending on the geographical origin. Here, we selected a specific cultivar, i.e., *Corylus Avellana*, and studied the effect of different chemical and thermal treatments (i.e., HTC, annealing at 900 °C, HTC plus annealing) on the final disordered carbon materials physico-chemical and electrochemical properties. Once the optimal hydrothermal and pyrolytic conditions had been identified, a further modification of the hard carbon was realized via laser irradiation in acetonitrile. The goal of this final process is to enrich the surface of the carbonaceous particles with nitrogen-containing groups. Disordered carbons functionalized with heteroatoms show increased capacity, surface wettability, and electronic conductivity [16]. In particular, functional groups containing nitrogen can alter Li absorption during discharge/charge process being N more electronegative than C [17,18,19].

Laser irradiation of particles in liquid environment is a powerful technique applicable to a wide range of materials for obtaining particles with desired morphological and compositional features, by tuning irradiation parameters, such as wavelength, energy, and pulse time of the laser, liquid composition, and duration of the experiment. A laser beam interacts with particles (both nanometric end/or micrometric) dispersed in a liquid, inducing chemical and morphological modification on the surface by a variety of different mechanisms [20]. When a laser beam is steered onto a liquid dispersion of particles, in fact, it can cause laser fragmentation in liquid (LFL) and/or laser melting in liquid (LML) [20,21,22,23]. LFL generates particles with reduced size; while, during LML, the photothermal melting of particles causes their fusion, thus the obtaining larger particles, or their vaporization, thus the reducing particles dimensions [20,21,22,23]. Moreover, by using reactive solvents, it is possible to obtain a chemical modification of surfaces. In this case the laser irradiation promotes a reaction between particles and solvent molecules, the so-called reactive laser irradiation (RLI) [22]. Here, for the first time in the literature as far as we know, a bio-waste derived hard carbon has been processed with laser irradiation to improve its electrochemical properties in LIBs by exploiting simultaneous LFL, LML and RLI.

All of the produced carbonaceous materials have been characterized with a multi techniques approach: morphology has been investigated by scanning and transmission electron microscopy as well as N_2_ absorption; crystallinity has been checked by X-ray diffraction and Raman spectroscopy; the chemical features of the surfaces have been investigated FT-IR and X-ray photoelectron spectroscopy; electrochemical properties in Li-ion half cells have been analyzed by galvanostatic techniques.

## 2. Materials and Methods

Hazelnut shells have been pulverized by the means of mechanochemical processes for 30 min using a M400 SpexShaker (SPEX SamplePrep, Metuchen, NJ, USA). Pulverization has been carried out in stainless steel jars, with 7 mm diameter stainless steel balls and a powder-to-balls weight ratio of 1:6. Milling sessions have been intermittently carried out at room temperature for 15 min followed by a 15 min rest to avoid the thermal deterioration of the samples. Starting from the pulverized bio-waste different chemical or thermal treatments have been carried out as summarized in the Table 1.

BIO04 was further modified by laser irradiation in liquid to obtain BIO04i. A sum of 0.330 mg of BIO04 was suspended in 15 mL of acetonitrile and magnetic stirred; for the irradiation a frequency-doubled Nd:YAG laser source (HandiYAG, Quanta System, Varese, Italy) (λ = 532 nm, τ = 7 ns, repetition rate = 10 Hz, fluence = 12 Jcm^−2^) was used. After 1 h 15’ of irradiation, the sample was dried under ambient condition.

Materials morphology and elementary composition were studied by the means of scanning electron microscope (SEM) (FEI ESEM XL30, Philips, North Billerica, MA, USA) coupled with an X-EDS EDAX microanalysis

Surface areas were measured by N_2_ adsorption using a Monosorb Quantachrome (Boynton Beach, FL, USA) apparatus and applying the Brauner, Emmet & Teller (BET) method.

X-ray diffraction (XRD) was performed using a X-Perth-Pro (Philips) diffractometer in the following conditions: CuKα radiation (λ = 1.5405600 Å), 2θ = 20°–60°, step size 0.040°, time per step 4 s.

FT-IR spectroscopy was carried by using a Jasco FT-IR 460 Plus (Tokyo, Japan) interferometer in the range 4000–400 cm^−1^. For Raman analysis, a DilorLabram micro-Raman (Horiba Jobin Yvon, Edison, USA) spectrophotometer were used (632 nm excitation wavelength, Peltier CCD camera detector, 1800 gr mm^−1^ grating, edge filter with a cut-off at 150 cm^−1^). Raman peaks fitting was carried out by using Voigt function with OriginPro 2016 software (OriginLab Corporation, Northampton, MA, USA).

XPS spectra were acquired by a Phoibos 100-MCD5 (SPECS GmBH, Berlin, Germany) spectrometer, using non-monochromatized AlKα radiation (1486.6 eV) operating at 10 kV and 10 mA. Wide spectrum was collected in Fixed Analyzer Transmission (FAT) mode with a pass energy of 50 eV and channel widths of 1.0 eV.

For transmission electron microscopy (TEM) BIO04 and BIO04i suspended in acetonitrile were dropped on holey carbon copper grids and observed with a Fei-TEC-NAI G2 20 TWIN (FEI Company, Hillsboro, OR, USA).

Electrochemical properties were studied by galvanostatic techniques (GC). Carbon electrodes were prepared via casting followed by drying under vacuum. Slurries were obtained by mixing carbonaceous active material (80%wt), Kynar 2801PVDF-polyvinylidene fluoride (10%wt) and Timcal Super P Carbon (10%wt) in tetrahydrofuran (THF). The slurry was casted on copper foils, dried at room temperature and cut in disks of 1 cm diameter. After drying under a vacuum at 120 °C for 4 h in a Buchi glass oven, electrodes were transferred in an ItecoEng SGS30 Ar-filled glove box with a moisture content below 0.1 ppm, where aprotic electrochemical cells were assembled. EL-CELL Std cells were used as test cells, carbon electrodes were coupled with Whatman glass fiber separators and lithium disk counter electrodes (Sigma Aldrich, Saint Louis, MO, USA). A 1 molal solution of LiPF6 dissolved in an ethylene carbonate-dimethyl carbonate solvent blend (1:1 in volume) has been used as electrolyte (Solvionic, Toulouse, France). GC experiments were performed using an 8-channel MTI battery test system, cycling in a 0–3 V range at 10/100/300 mAg^−1^ current rates. Electrochemical impedance spectroscopy tests (EIS) have been performed using an Ivium Vertex instrument by applying a sinusoidal ΔV = 10 mV in the 200 kHz–0.1 Hz frequency range in a 3 electrodes configuration.

## 3. Results and Discussion

### 3.1. Physico-Chemical Characterization of the Carbon Preparation Process

SEM micrographs were recorded on all of the samples (Figure 1) to examine the morphology of the different carbonaceous particles. BIO01 shows smooth surface whereas BIO02-03-04 have similar morphologies even below the microscale: in all of these samples pores and granular structures can be observed. On the other hand, surface area measured by BET gives 0.3, 47.3, 96 and 382 m^2^/g for BIO01, BIO02, BIO03, and BIO04, respectively. Both hydrothermal and annealing treatments lead to an increase in the surface area, the largest values obtained being BIO04, where both the processes were applied.

The increase in surface area, especially in the case of BIO03 and BIO04 samples, is probably due to the formation of micropores during the pyrolysis because of gas evolution [24], rearrangement of carbon atoms and the agglomeration of nanostructures [5,25].

As already mentioned, hazelnut shells are mainly constituted by lignin, cellulose and hemicellulose [26]. During the pyrolysis at a high temperature under inert/reducing atmospheres, these 3 components degrade in different temperature ranges; hemicellulose decomposes between 220 °C and 315 °C, cellulose degrades in the 315–440 °C range, whereas lignin degradation starts at 220 °C ending at 800 °C [27]. FTIR spectroscopy highlights easily the presence/absence of lignin/cellulose/hemicellulose bands [28]: in the Figure 2, a comparison between FTIR spectra of all the samples is reported, while peaks assignment is summarized in Table 2.

Samples BIO01 and BIO02 show bands in the FTIR spectra easily assigned to hemicellulose/cellulose/lignin [28]. None of H, C or L signals are visible in the FTIR spectra of BIO03 and BIO04 due to the complete degradation at 900 °C, in line with literature [24,27]. Furthermore, both pyrolyzed carbons (BIO03 and BIO04) do not show any band in the range 3000–3800 cm^−1^ attributable to -OH stretching, nor others related to C=O and C-O vibrational modes in the 1200–1750 cm^−1^ range, thus suggesting the almost complete elimination of oxygen at high temperature. Oxygen depletion has been confirmed by EDX measurements: the O/C atomic ratio are of 0.4, 0.3, 0.06, and 0.04 for BIO01, BIO02, BIO03, and BIO04, respectively. This trend aligns with the expected pyrolysis mechanism as at 900 °C in inert condition oxygen moieties are lost via CO, CO_2_, H_2_O and other byproducts [24,29]. XRD patterns all samples are shown in the Figure 3a and Raman spectra of carbons before and after pyrolysis are shown in the Figure 3b–d.

Biomass-derived materials constituted by cellulose, lignin and hemicellulose are poorly detectable by XRD due to their amorphous state [30,31]. In the pyrolyzed samples BIO03-BIO04,2 broad reflections typical of amorphous carbons with nanocrystalline domains at about 2θ = 23°and 43°can be observed: both are related to a graphite-like structural arrangement. In particular, the first peak is related to (002) interlayer reflection of stacked aromatic layers, whereas the second peak is due to in-layer (100) reflection [5,11,25].

Overall, as expected, pyrolysis drives an evolution of the biomass to a poorly graphitized material. BIO04 pattern shows (002) peak at larger diffraction angles, thus suggesting a higher graphitization degree compared to BIO03 [24,32]. Therefore, the hydrothermal treatment facilitates the graphitization at high temperature during the annealing. This evidence alignswith the Raman spectroscopy results where the appearance of typical D and G bands are observed in BIO03 and BIO04 whereas for BIO01only peaks related to a lignocellulosic material are visible (Table 3). On passing we can mention that for BIO02 a strong photoluminescence has been detected confirming the structural degradation of the biomass during the hydrothermal treatment. Looking more in detail in the Raman spectra of the pyrolyzed samples, the carbon peaks between 1000 and 1600 cm^−1^ can be deconvoluted in 5components D4, D3, D2, D1, and G, following the widely accepted model of Sadezky [33]. G peak at ~1595 cm^−1^ is related to graphitic sp^2^ carbon with E_2_g symmetry, D1 at ~1345 cm^−1^ is related to vibration mode of microcrystallite graphite with A1g symmetry, D3 at ~1500 cm^−1^ is attributable to amorphous carbon contribution, D2 (~1630 cm^−1^) and D4 (~1200 cm^−1^) are related to disordered graphitic lattice (sp^3^-rich phase), with A_1_g and E_2_g symmetry, respectively [24,33,34]. In the Raman spectra of amorphous carbon, as suggested by Ferrari and Robertson [35], a G peak position of up to 1590 cm^−1^ indicates the presence of nanocristalline graphite in carbonaceous material. The I_G_/I_D3_ peak area ratio, used as indicator of graphitization degrees [24], is of 0.99 and 1.27 for BIO03 and BIO04, respectively, confirming the results obtained with XRD. Moreover, I_D1_/(I_D1_ +I_G_) ratio, which is an indicator of defect concentration, is close to 0.6 for both materials (0.62 and 0.56 for BIO03 and BIO04, respectively) suggesting an extensive defectivity, both in layer and out-layer, of both hard carbons.

In summary, BIO04 shows the largest surface area, smaller O/C ratio, and largest graphitization degree with respect to our other carbonaceous materials. Considering the different preparation routes of BIO03 and BIO04, the morphological and structural differences between the two carbonaceous materials originate by the hydrothermal pre-treatment. BIO04 has been therefore selected as the starting material for evaluating the impact of irradiation with laser on the physicochemical and electrochemical properties of this hard carbon.

### 3.2. Impact of the Laser Irradiation on the BIO04i Physicochemical Properties

The BIO04i material has been obtained by nanosecond pulsed laser irradiation of BIO04 resuspended in acetonitrile. As above mentioned, laser irradiation can induce morphological, structural, and compositional modification.

SEM micrographs, shown in Figure 4, demonstrate the effect of laser irradiation on the carbon particles morphology: apparently a massive formation of sponge-like morphologies is observed. This remarkable change leads to a huge increase in the surface area to 760 m^2^/g, doubling the BIO04 value.

In the Figure 5 TEM images of BIO04 and BIO04i are shown.BIO04 consists of tens of nanometers-sized particles, whereas in the BIO04i sample particles are strongly modified [20,22,37]. Apparently, the irradiated particles are constituted by a composite morphology where large agglomerates of bent graphitic layers (see Figure 5d) are surrounded by more compact amorphous carbon regions. At larger magnifications (Figure 5e) it is possible to highlight the stacking of few graphitic nano-sheets with average interlayer spacing of 0.33 (±0.07) nm corresponding to d_002_ graphite interlayer spacing [32,38].

From XRD patterns (Figure 6a) no substantial differences can be detected between the BIO04 and BIO04i.

The graphitization degree evaluated by the IG/ID3 Raman peaks ratio, is almost unaltered after irradiation, whereas the defect concentration strongly decrease for BIO04i being the ID1/(ID1 + IG) 0.17 in respect of the 0.56 value of BIO04.

In the FTIR spectra (Figure 6d), the BIO04i material shows 2distinctive signals at 1630 and 3450 cm^−1^, undetectable in BIO04: these bands can be attributed to C = N bending and N-H stretching, respectively, indicating that laser irradiation in acetonitrile induced the implantation of nitrogen moieties on the graphene sheets. Remarkably, the lack of a strong absorption atabout2225 cm^−1^suggests the absence of C≡N groups bonded over the carbon particles [38].

The implantation of nitrogen moieties on the carbon surface has been confirmed by XPS as shown in the Figure 6e. The N1s signal has been detected only in BIO04i with a N/C area ratio of 0.008 (corresponding to an estimated N content on the surface of 0.6–0.7%wt), whereas it is absent in the BIO04 one. The N 1s peak position at 399.86 eV indicates a pyrrolic or pyridonic nitrogen character, in line with FT-IR results. In fact, pyrrolic nitrogen is bonded to 2 carbons in a 5-memebered ring, contributing with two electrons to the π system, while pyridonic nitrogen is in a 6-membered ring with an oxygen functionality [38,39].

The implantation of functional groups containing nitrogen originates from the laser irradiation of the suspension, where the solvent and the carbon particles interact with the laser beam. The irradiated focus can reach local temperatures large enough to melt the carbon particles and to decompose the solvent molecules. As discussed by Jung et al. in the case of acetonitrile the laser irradiation leads to the formation of CH_3_ and CN radicals/ions as well as C, H, and N monatomic species/ions [40]. It is well known that HCN is rapidly formed and dissolved in acetonitrile by the reaction between CN radical and H ions [41], while N ions can be available for a fast interaction with C and H over the surface of the melted carbon particles.

Moreover O/C area ratio, that was of 0.07 and 0.08 for BIO04 and BIO04i, respectively, indicating a low degree of carbon oxidation even after laser irradiation, useful for battery application to limit SEI formation [5,42].

### 3.3. Impact of the Laser Irradiation on the Electrochemical Performance of the Hard Carbon in a Lithium Cell

In view of the remarkable alteration of the morphology, structure and surface moieties obtained by laser irradiation, both the hard carbon materials, i.e., BIO04 and BIO04i, have been studied as active materials in negative electrodes for secondary Li-ion batteries. As far as we know, this is the first ever reported demonstration of the use of laser irradiation to enhance the electrochemical performance of an electrode material in aprotic batteries.

The electrochemical performances of BIO04 and BIO04i in lithium half cells have been evaluated by galvanostatic cycling at 10 mAg^−1^ and the experimental results are shown in the Figure 7. The performance of the irradiated hard carbon material is remarkably superior compared to the pristine one.

The BIO04 hard carbon shows a reversible electrochemical activity in the lithium cells in line with other hard carbons from biomasses [8,13]. However, similarly to many other hard carbons, the excellent specific capacity exchange in the first cycles, largely beyond the theoretical capacity of graphite (i.e., 372 mAhg^−1^), is counterbalanced by the huge irreversible capacity loss in the first cycle (i.e., ~700 mAhg^−1^ corresponding to a coulombic efficiency of 44%) and the unsatisfactory capacity retention, being the specific capacities measured during de-lithiation at cycles 1, 10 and 20, respectively, 530, 293 and 221 mAhg^−1^ (with a capacity retention at cycle 20 of 42% in respect to cycle 1).

Turning to the irradiated material, the BIO04i electrodes show an extraordinary increase in the performance in terms of specific capacity exchanged in de-lithiation (1108, 682 and 578 mAhg^−1^at cycles 1, 10 and 20, respectively), capacity retention (52% at cycle 20 in respect to cycle 1) and coulombic efficiency in the first cycle (56%). Moreover, the mean working potential is reduced to 0.45 V vs. Li with respect to the 0.5 V vs. Li of the BIO04 material, thus leading to a direct increase in the energy density of a Li-ion configuration in respect to any positive electrode.

This enhancement of the performance originates from the irradiation of the hard carbon and is likely due to the beneficial effect of two cooperating effects: (a) the presence of nitrogen-containing moieties and, (b) the expansion of the specific surface area [43,44]. The evidence reported here does not enable a clear decoupling of their individual corresponding impact. We can speculate about the possible synergy to boost the efficiency of the Li incorporation/de-incorporation mechanism of BIO04i compared to the BIO04 sample [18,38,39].

Overall, both BIO04 and BIO04i consist in porous disordered carbon and graphite crystallites. Besides graphene-inter-layer intercalation, the occurrence of a “micropore ionic couple storage mechanism” is well accepted for disordered carbons in the literature [45,46,47]. In fact, upon reduction, Li^+^ can chemisorb in the micro- and nano-pores forming C/Li clusters, going beyond the LiC_6_ stoichiometry expected from the graphite intercalation. This mechanism has been proved to be highly reversible in oxidation and highly efficient for hundreds of cycles [48,49]. In this respect, porosities provide active sites for lithium storage, especially in the irradiated carbon material, where we observe a remarkable sponge-like morphology. In addition, nitrogen surface moieties possibly generate additional facile chemisorption sites by improving the electronegativity of neighboring carbons supporting the formation of (Li^+^ e^-^) pairs.

A remarkable outcome of laser irradiation is the increase in the coulombic in the first cycle: irreversible capacity losses are common in carbon electrodes due to the degradation of the electrolyte and the release of gaseous species [4,5,6]. In disordered carbons this parasitic chemistry is typically enhanced by the presence of residual moieties over the carbon surface after pyrolysis [5,48]. Apparently, the use of laser irradiation mitigates the irreversible charge loss by increasing the coulombic efficiency from 44 to 56%: these values are unsatisfactory for practical applications, but improvements are possible. In cat, the optimization of electrolyte additives and composition can strongly decrease the irreversible capacity as well as the optimization of an electrochemical cycling protocol for formation.

An operando or ex situ characterization of the BIO04i samples are surely necessary to evaluate this material as a candidate negative electrode for Li-ion batteries and to decouple the effect of N-implantation and surface area increase. However, this goal is beyond the scope of this work. Our aim is to prove the beneficial impact of irradiation in solvent on the physicochemical and electrochemical properties of hard carbon to tune the morphology and surface moieties. The here reported experimental evidence unequivocally prove this point.

To extend the analysis of the performance of BIO04 and BIO04i electrodes, prolonged cycling at higher current rates have been performed as shown in the Figure 8a. Moreover, impedance spectra of the working electrodes (BIO04 and BIO04i) after cycling are shown in the Figure 8b to highlight the impact of irradiation on the stability of the dielectric properties upon cycling.

BIO04i outperforms BIO04 at 100 and 300 mAg^−1^ for the entire cycling thus proving the beneficial effect of the irradiation on the performance in lithium half cells. The enhancement of the performance is likely related to the improved dielectric properties of the BIO04i electrode compared to BIO04 as demonstrated in the Figure 8b by EIS. Impedance spectra have been fitted with an equivalent circuit shown in the Figure 8b: optimized parameters are listed in the figure caption.

The electrode impedance spectra after 110 galvanostatic cycles is 50% smaller for the irradiated samples compared to BIO04. In particular, in the EIS 2 semicircles can be identified; the low-frequency semicircleis likely related to the charge transfer step, in consideration of the corresponding time constant 7.5 ms, and the high frequency semicircle originated by the passivation film. Both semicircles shrink on irradiated electrodes thus suggesting improved passivation films and charge transfer kinetics. Considering this, one can speculate that the larger surface area in the BIO04i samples compared to the BIO04 enhances the capacitance of both passivation film and electrode double layer. On the other hand, the nitrogen moieties possibly promote the precipitation of a thinner, and less resistive, passivation film and facilitate the charge transfer. Overall, EIS supports our hypothesis of a cooperative beneficial effect provided by irradiation thorough increase in surface area and implantation of nitrogen moieties.

It is important to underline that many open fundamental questions and performance problems still need to be addressed, in particular regarding a detailed analysis of the origin of the enhancement of the lithium storage ability as well as valuable mitigation strategies to limit the irreversible capacity losses. Further studies are currently in progress in our laboratories to investigate these topics.

## 4. Conclusions

Pyrolyzed carbon from bio-waste sources was studied for use as an anodic material for Li-ion batteries. The combined effect of hydrothermal treatments and annealing at 900 °C allows to obtain a carbon material with high surface area and low O/C ratio. A tailored laser irradiation step in acetonitrile of the obtained hard carbon leads to a further increase in the surface area, reduction in the graphene defectivity and implementation of nitrogen moieties on the carbon surface. Once tested in aprotic secondary batteries the irradiated hard carbon material can largely overcome the performance of the unirradiated hard carbon as well as pyrolytic graphite.

## Figures and Tables

**Figure 1 nanomaterials-11-03183-f001:**
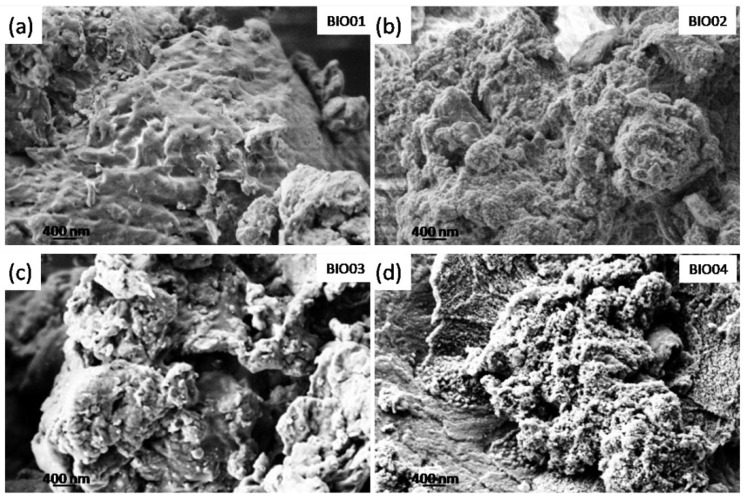
SEM images of (**a**) BIO01; (**b**) BIO02; (**c**) BIO03 and (**d**) BIO04.

**Figure 2 nanomaterials-11-03183-f002:**
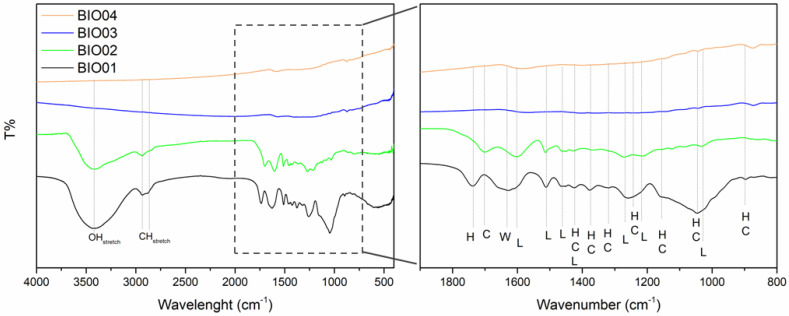
FTIR spectra of BIO01, BIO02, BIO03, and BIO04, where H, C, L and W stand for hemicellulose, cellulose, lignin, and water, respectively.

**Figure 3 nanomaterials-11-03183-f003:**
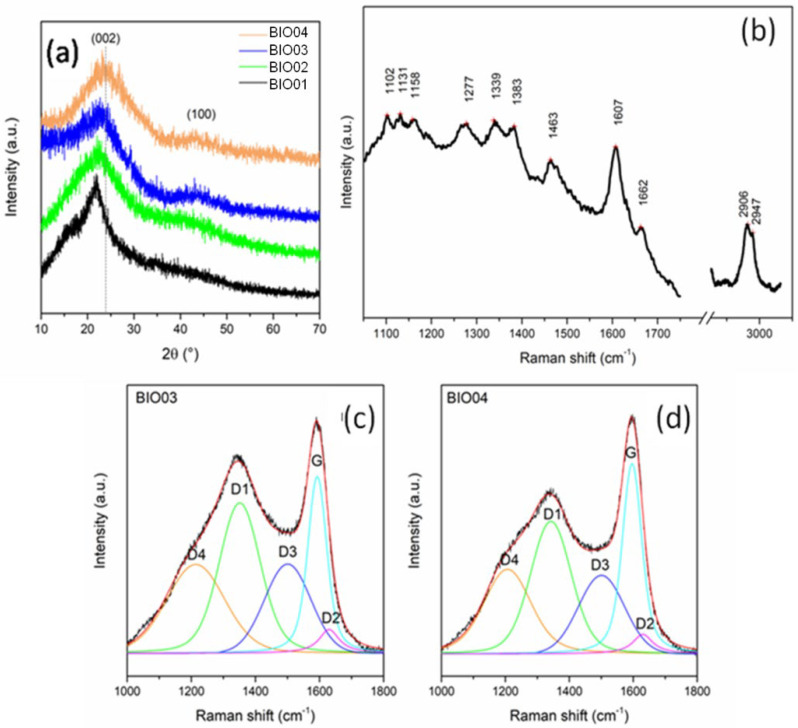
(**a**) Normalized XRD pattern of BIO01, BIO02, BIO03, and BIO04; (**b**) Raman spectra of BIO01; and curve-fitted Raman spectra of (**c**) BIO03 and (**d**) BIO04.

**Figure 4 nanomaterials-11-03183-f004:**
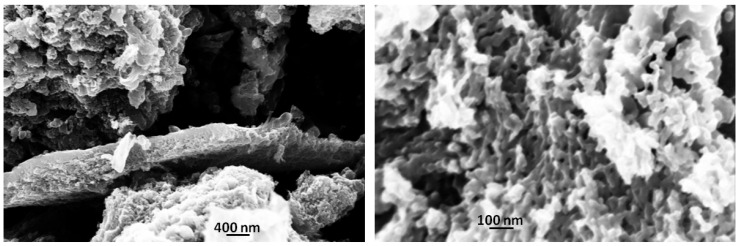
SEM images of BIO04i at different magnification.

**Figure 5 nanomaterials-11-03183-f005:**
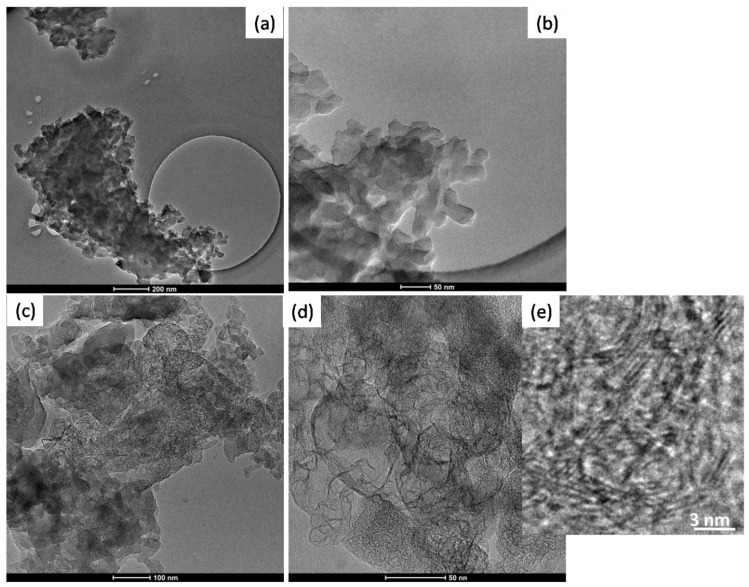
TEM images of (**a**,**b**) BIO04 and (**c**–**e**) BIO04i at different magnifications.

**Figure 6 nanomaterials-11-03183-f006:**
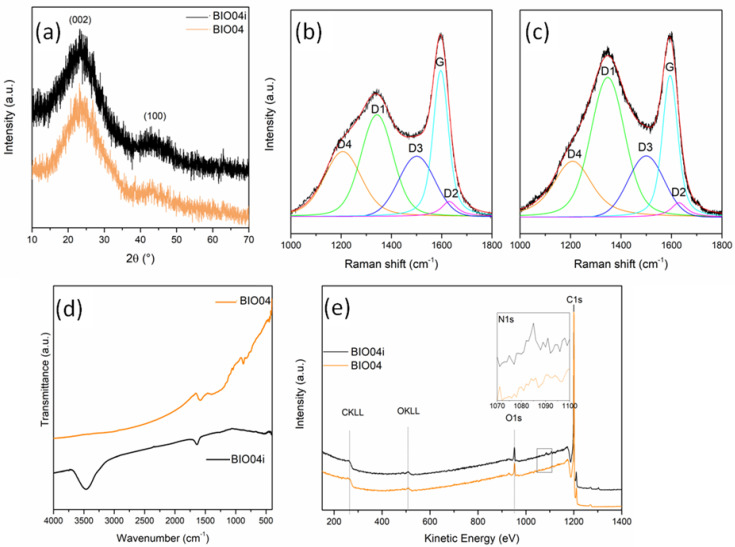
(**a**) XRD patterns of BIO04 and BIO04i; curve-fitted Raman spectra of (**b**) BIO04 and (**c**) BIO04i; (**d**) FT-IR spectra of BIO04 and BIO04i; and (**e**) XPS wide spectra of BIO04 and BIO04i.

**Figure 7 nanomaterials-11-03183-f007:**
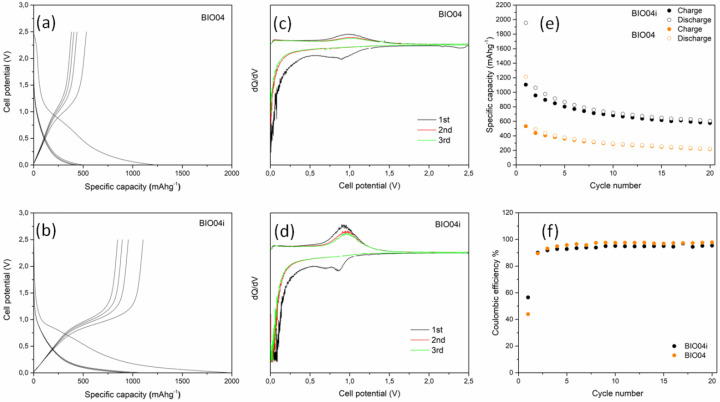
Voltage profile for the first cycles of (**a**) BIO04 and (**b**) BIO04i; Capacity derivative vs. voltage for the first 3 cycles of (**c**) BIO04 and (**d**) BIO04i; galvanostatic cycling performances in terms of specific capacity (**e**) and coulombic efficiency (**f**) of electrodes cycled at 10 mAg^−1^.

**Figure 8 nanomaterials-11-03183-f008:**
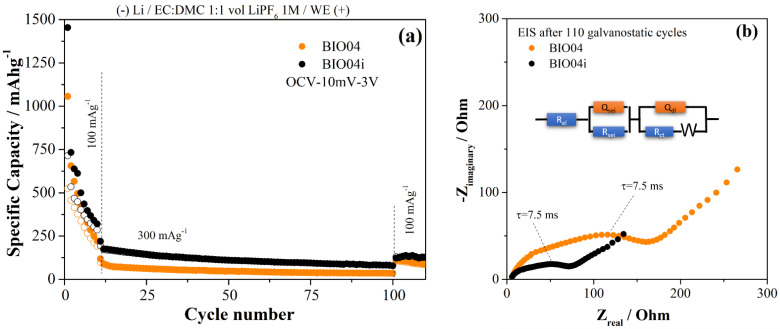
(**a**) comparison of the performance in galvanostatic tests of the BIO04 and BIO04i samples; (**b**) EIS spectra recorded on BIO04 and BIO04i electrodes after cycling and equivalent circuit adopted for modeling (BIO04: RSEI = 35 Ω; RCT = 140 Ω; QSEI = 1.1 μF; QCT = 54 μF; BIO04I: RSEI = 22 Ω, RCT = 55 Ω; QSEI = 1.8 μF; QCT = 140 μF).

**Table 1 nanomaterials-11-03183-t001:** Sample labels and preparation processes.

Code	Starting Materials	Preparation	Pyrolysis
BIO01	hazelnut shells	Pulverization by ball milling	None
BIO02	BIO01 2 g, Citric acid 25 mg, H_2_O 5 g	Mixing in ultrasonic bath in Teflon liner, hydrothermal treatment in autoclave at 220° for 12 h, drying at 80° for 6 h.	None
BIO03	BIO01	None	Annealing under Ar/H_2_ 5% flow (120 mL/min) (3 h at 20°, heating 5°/min, 2 h at 900°, natural cooling)
BIO04	BIO02	None	

**Table 2 nanomaterials-11-03183-t002:** FTIR assignment in the region 800–2000 cm^−1^.

Band Peak(cm^−1^)	Species	Assignment [5,28]
896	H/C	Anometric vibration at β-glycosidic linkage
1028	L	C-O stretching
1045	H/C	C-OH stretching
1156	H/C	C-O-C antisymmetric stretching
1219	L	C-O stretching (esters or ethers)
1240	H/C	C-O stretching (carboxylic acid)
1265	L	C-O stretching
1316	H/C	O-H in-plane bending
1375	H/C	C-H bending
1423	L	C-H bending (C-H_3_)
1462	L	C-H bending (C-H_2_)
1510	L	Aromatic skeletal vibration
1600	L	Aromatic skeletal vibration
1636	Adsorbed Water	O-H bending
1702	C	C=O stretching
1735	H/L	C=O stretching

where H, C and L stand for hemicellulose, cellulose, and lignin, respectively.

**Table 3 nanomaterials-11-03183-t003:** Raman bands assignment of BIO01.

Bands Position (cm^−1^)	Species	Assignment [36]
1102	C/H	Stretching C-C, C-O
1131	C/H	Stretching C-C, C-O
1158	L	Stretching C-C, C-O and HCC bending
1277	L/C	aryl-O, aryl-OH
1339	C	HCC and HCO bending
1383	L/C	HCC and HCO bending
1463	L	HCH and HCO bending
1607	L	Stretching aromatic ring
1662	C/H	C=C of coniferyl alcohol and coniferaldehyde
2906	L/C/H	CH stretching
2947	L/C/H	CH stretching

where H, C and L stand for hemicellulose, cellulose and lignin, respectively.

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
