# Peer review of "Laser Irradiation of a Bio-Waste Derived Carbon Unlocks Performance Enhancement in Secondary Lithium Batteries"

_nanomaterials, 2021, doi:10.3390/nano11123183_

Round 1
Reviewer 1 Report
This manuscript repots the synthesis of the hazelnut shell-derived carbon with large specific surface area and nitrogen-containing moieties by laser irradiation as anode materials for lithium-ion battery. The bio-waste sources can be recycled and the electrode exhibits high capacity. Considering this, it is suitable for publication after Major Revisions. And the authors should also carefully consider the following suggestions:
- The specific capacity of BIO04 and BIO04 is far beyond the theoretical capacity of graphite. Please explain the lithium-ion storage mechanism of BIO04 and BIO04 in detail.
- The initial Coulomb efficiency of BIO04 and BIO04 is low. Please explain the reason and discuss how to improve the initial Coulomb efficiency.
- It is better to fit the EIS to obtain the charge transfer impedance of BIO04 and BIO04i to support the statement that “the nitrogen moieties possibly promote the precipitation of a thinner, and less resistive passivation film and facilitate the charge transfer”.
- Some papers are related to this work and maybe helpful to the readers. (J. Phys. Chem. C 2021, 125, 19060; Carbon Energy. 2019, 1(1), 13; Carbon Energy. 2019, 1(2), 253)
Author Response
This manuscript repots the synthesis of the hazelnut shell-derived carbon with large specific surface area and nitrogen-containing moieties by laser irradiation as anode materials for lithium-ion battery. The bio-waste sources can be recycled and the electrode exhibits high capacity. Considering this, it is suitable for publication after Major Revisions. And the authors should also carefully consider the following suggestions:
1. The specific capacity of BIO04 and BIO04 is far beyond the theoretical capacity of graphite. Please explain the lithium-ion storage mechanism of BIO04 and BIO04 in detail.
Reply: We thank the reviewer for this comment. Both BIO04 and BIO04i consist in porous disordered carbon and graphite crystallites. Besides graphene-inter-layer intercalation, the occurrence of a “micropore ionic couple storage mechanism” is well accepted.1-3 Upon reduction Li+ chemical adsorbs and intercalates in the micro and nanopores forming C/Li clusters, going beyond the LiC6 stoichiometry expected from the graphite intercalation. This mechanism has been proved to be highly reversible in oxidation and highly efficient for hundreds of cycles.4-5 In this respect, extended porosities provide active sites for lithium storage, especially in the irradiated carbon material, where we observe a remarkable sponge-like morphology. In addition, nitrogen surface moieties possibly generate additional facile chemisorption sites by improving the electronegativity of their neighboring carbons energetically accessible for ( Li+ e‾ ) pairs. We updated the manuscript by reporting an appropriate content in the section 3.3 with appropriate additional references.
1) Dhan, J.R.; Zheng, T.; Xue, J.S. Mechanism for lithium insertion in carbonaceous materials. Science 1995, 270, 590-593
2) Wang, S.; Matsui, H.; Tamanura, H.; Matsumura Y.; Yamabe, T. Mechanism of lithium insertion into disordered carbon. Phys. Rev. B 1998, 58, 8163
3) Wang S.; Kakumoto, T.; Matsui, H.; Matsumura, Y. Mechanism of lithium insertion into disordered carbon. Synthetic Metals 1999, 103, 2523-2524
4) Bonino, F. Brutti, S.; Reale, P.; Scrosati B.; Gherghel L.; Wu, J.; Mullen, K. A disordered carbon as a novel anode materials in lithium-ion cells. Advanced Materials 2005, 17, 743-746
5) Togonon, J.J.H.; Chiang, P.-C.; Lin, H.-J.; Tsai, W.-C.; Yen, H.-J. Pure carbon-based electrodes for metal-ion batteries. Carbon trends 2021, 3, 100035
2. The initial Coulomb efficiency of BIO04 and BIO04 is low. Please explain the reason and discuss how to improve the initial Coulomb efficiency.
Reply: We thank the reviewer for this comment. The irreversible capacity loss is common for carbon-based materials and can be due to the initial formation of SEI related to the decomposition of electrolyte, in addition the disordered carbon structure may favor the irreversible capacity by trapping lithium ions. We extended the related discussion in the text also providing possible research lines to further enhance the reversibility of the electrochemical reactions.
3. It is better to fit the EIS to obtain the charge transfer impedance of BIO04 and BIO04i to support the statement that “the nitrogen moieties possibly promote the precipitation of a thinner, and less resistive passivation film and facilitate the charge transfer”.
Reply: We thank the reviewer for this comment. We fitted the EIS data in the figure 8 using an equivalent circuit to further support our discussion. We added these new data in the text with appropriate comments.
4. Some papers are related to this work and maybe helpful to the readers. (J. Phys. Chem. C 2021, 125, 19060; Carbon Energy. 2019, 1(1), 13; Carbon Energy. 2019, 1(2), 253)
Reply: We thank the reviewer for the suggestion, we added these three references in the revised text.
Reviewer 2 Report
The utilization of laser irradiation is very interesting, furthermore converting bio-waste to battery material shows good potential. However, the only concern is claiming the material has superior performance for Li-ion batteries. With high average voltage, 0.45V, high ICE, and 250 mAh/g with 100 mA/g provide very low performance compared to graphite. The performance is not even close to the graphite and it is difficult to agree the material has superior performance. I suggest re-evaluating the tile of the manuscript.
Author Response
We thank the reviewer for this comment: it is true that performance are far from being optimized and suitable for exploitation. We revised the title to avoid any misunderstanding.
We thank also the reviewer for the positive overall evaluation.
Reviewer 3 Report
The paper describes the effect of laser irradiation in acetonitrile of pyrolized carbons obtained from bio-waste sources on their electrochemical properties. The authors have shown that the performance of irradiated materials as anodes in lithium ion batteries is clearly superior to that of unirradiated carbons. Undoubtedly, the article focuses on a topic currently of enormous interest. Moreover the work has been well planned and executed being experiments and results described in detail. Therefore I consider that the article will be interesting for the general reader of this publication.
The authors suggest that the improvement in the properties of the irradiated materials may be due both to the presence of nitrogen and to the increase in surface area. However, the nitrogen content of the irradiated sample is not included in the article, and this is a fundamental parameter to to compare the properties of these carbons with those of other analogues enriched in nitrogen by other routes. I consider that the nitrogen content should be determined and, depending on it, its possible influence on the improvement of the electrochemical properties should be commented.
Further comment: I recommend including the Raman spectrum of the BIO04 sample in figure 6b.
Author Response
The paper describes the effect of laser irradiation in acetonitrile of pyrolized carbons obtained from bio-waste sources on their electrochemical properties. The authors have shown that the performance of irradiated materials as anodes in lithium ion batteries is clearly superior to that of unirradiated carbons. Undoubtedly, the article focuses on a topic currently of enormous interest. Moreover the work has been well planned and executed being experiments and results described in detail. Therefore I consider that the article will be interesting for the general reader of this publication.
Reply: We would like to thank the reviewer for the positive overall evaluation of our manuscript.
The authors suggest that the improvement in the properties of the irradiated materials may be due both to the presence of nitrogen and to the increase in surface area. However, the nitrogen content of the irradiated sample is not included in the article, and this is a fundamental parameter to to compare the properties of these carbons with those of other analogues enriched in nitrogen by other routes. I consider that the nitrogen content should be determined and, depending on it, its possible influence on the improvement of the electrochemical properties should be commented.
Reply: We would like to thank the reviewer for this suggestion. The nitrogen content estimated by XPS is reported in the text (line 289)
Further comment: I recommend including the Raman spectrum of the BIO04 sample in figure 6b
Reply: We would like to thank the reviewer for this suggestion. We included Raman spectrum of the BIO04 in figure 6b and caption has been rewritten (rows 272-274)
Reviewer 4 Report
Please see attached file for comments.

Author Response
Dear Authors, The ideas in the manuscript are interesting. As you point out there is still a long way to go in terms of useful applications of these bio derived materials The manuscript needs to be improved slightly. I suggest reading through the manuscript properly and polishing the English.
Reply: we would like to thank the reviewer for the positive feedback and all suggestions.
I have pointed out a few observations.
- Line 44, seems to be missing some information or the sentence structure is incorrect. Please review.
Reply: We revised the sentence accordingly to the reviewer comment
- Line 49, please check tense.
Reply: We checked the quality of the english and revised the sentence.
- Line 74, please check sentence construction.
Reply: We revised the sentence accordingly to the reviewer comment
- Line 79, please check sentence construction.
Reply: We revised the sentence accordingly to the reviewer comment
- Line 92, please consider reframing sentences to shorter sentences for better clarity.
Reply: We revised the sentence accordingly to the reviewer comment
- Line 110-117, please check the tense. Some sentences are in past tense while some are not.
Reply: We revised the sentence accordingly to the reviewer comment
- Section 3.1- Please rearrange to explain a little about the difference in morphology before proceeding to talk about surface area.
Reply: We revised the sentence accordingly to the reviewer comment
- Fig 3(a) 100 peak can be seen for BIO02 as well. Can you please explain this?
Reply: we thank the reviewer for this comment. The (100) peak is related to the in-layer ordering along the aromatic sheets; HTC induces the partial degradation of main hazelnut shells polymers, during which the rearrangement of aromatic structures can occur.
- XRD peak shift may just mean larger crystallite size. Can you please provide some literature references to support your claim that XRD peak shift means higher graphitization?
Reply: we thank the reviewer for this comment. As reported in ref [33], graphitization process induces the reducing of the d002-spacing (hence the related diffraction peak is shifted at higher diffraction angles); as reported in ref [24] the decrease of the d002 value (obtained by XRD) is attributable to short-range order as a result of strong van der Waals interaction between adjacent graphene layers.
- I didn’t find any explanation for Fig 6(a), please include some explanation.
Reply: we thank the reviewer for this comment. we added appropriate comments to figure 6(a)
- Did you do any quantification of the amount of N2 on the surface?
Reply: we thank the reviewer for this comment. As mentioned above, we estimated by XPS the N/C ratio and evaluate the overall nitrogen doping on the surface. Comments have been reported in the text.
- Can you please explain why you coated this material on Al foil and not Cu foil for your electrochemical property evaluation?
Reply: we thank a lot the reviewer for this remark: we casted the material on copper foil, “Al” was a typo
Round 2
Reviewer 1 Report
Authors have revised the manuscript carefully and all the questions raised have been addressed well. Therefore I recommend it for publication at the present form.